

# Arabidopsis thaliana exudates induce growth and proteomic changes in Gluconacetobacter diazotrophicus

Tamires Cruz dos Santos*, Mariana Ramos Leandro*,
Clara Yohana Maia, Patrícia Rangel, Fabiano S. Soares, Ricardo Reis,
Lucas Passamani, Vanildo Silveira and Gonçalo
Apolinário de Souza Filho

Laboratório de Biotecnologia/Unidade de Biologia Integrativa, Universidade Estadual do Norte
Fluminense, Campos dos Goytacazes, Rio de Janeiro, Brazil
* These authors contributed equally to this work.

## ABSTRACT

**Background:** Plants interact with a variety of microorganisms during their life cycle, among which beneficial bacteria deserve special attention. *Gluconacetobacter diazotrophicus* is a beneficial bacterium able to fix nitrogen and promote plant growth. Despite its biotechnological potential, the mechanisms regulating the interaction between *G. diazotrophicus* and host plants remain unclear.

**Methods:** We analyzed the response of *G. diazotrophicus* to cocultivation with *Arabidopsis thaliana* seedlings. Bacterial growth in response to cocultivation and plant exudates was analyzed. Through comparative proteomic analysis, *G. diazotrophicus* proteins regulated during cocultivation were investigated. Finally, the role of some up-accumulated proteins in the response *G. diazotrophicus* to cocultivation was analyzed by reverse genetics, using insertion mutants.

**Results:** Our results revealed the induction of bacterial growth in response to cocultivation. Comparative proteomic analysis identified 450 bacterial proteins, with 39 up-accumulated, and 12 down-accumulated in response to cocultivation. Among the up-accumulated pathways, the metabolism of pentoses and protein synthesis were highlighted. Proteins potentially relevant to bacterial growth response such as ABC-F-Etta, ClpX, Zwf, MetE, AcnA, IlvC, and AccC were also increased. Reverse genetics analysis, using insertion mutants, revealed that the lack of ABC-F-Etta and AccC proteins severely affects *G. diazotrophicus* response to cocultivation. Our data demonstrated that specific mechanisms are activated in the bacterial response to plant exudates, indicating the essential role of "ribosomal activity" and "fatty acid biosynthesis" in such a process. This is the first study to demonstrate the participation of EttA and AccC proteins in plant-bacteria interactions, and open new perspectives for understanding the initial steps of such associations.

# INTRODUCTION

The interaction between bacteria and plants involves complex recognition and signaling mechanisms that activate specific responses (*Dennis, Miller & Hirsch, 2010*;

Corresponding author
Gonçalo Apolinário de Souza Filho,
goncalos@uenf.br

*Sasse, Martinoia & Northen, 2018*). Among plant-associated microorganisms, the importance of plant growth-promoting bacteria (PGPB) in agriculture has been increasing, reducing the demand for industrialized fertilizers (*Grobelak et al., 2018*; *Nkebiwe, Weinmann & Müller, 2016*). PGPB may interact with plants by colonizing the rhizosphere or the surfaces of leaves and roots or by establishing endophytic colonization (*Lery et al., 2008*; *Zúñiga et al., 2017*).

*Gluconacetobacter diazotrophicus* is an endophytic nitrogen-fixing PGPB that was isolated for the first time from sugarcane plants (*Cavalcante & Döbereiner, 1988*). This bacterium can colonize other plant species such as sweet potato, pineapple, coffee, and *Arabidopsis* (*Reis, Olivares & Döbereiner, 1994*; *Souza et al., 2016*). Among the relevant characteristics of this bacterium is its ability to colonize and promote the growth of both dicot and monocot plants (*Cavalcante & Döbereiner, 1988*; *Souza et al., 2016*). *G. diazotrophicus* is considered a bacterial model for the study of endophytic associations, providing a better understanding of the pathways involved in this process (*Bertalan et al., 2009*; *Lery et al., 2011*; *Meneses et al., 2017*).

*Souza et al. (2016)* showed that *G. diazotrophicus* establishes a positive endophytic association with *Arabidopsis thaliana* resulting in biomass gain and increased photosynthetic efficiency. In addition, the early stages of the association of *G. diazotrophicus* with *A. thaliana* appears to activate specific responses of the plant immune system, but even so *G. diazotrophicus* is able to colonize and promote plant growth (*Souza et al., 2016*). Although some studies have sought to elucidate the molecular mechanisms modulating the interaction between *G. diazotrophicus* and sugarcane, few studies have addressed such mechanisms in dicot host plants (*Lery et al., 2011*; *dos Santos et al., 2010*; *Meneses et al., 2011*).

Genomic, transcriptomic and proteomic studies have addressed the genes and proteins that are regulated during the association of *G. diazotrophicus* and its monocot plant hosts (*Lery et al., 2011*; *Meneses et al., 2017*; *dos Santos et al., 2010*; *Meneses et al., 2011*). Proteomic approaches have been used to analyze the main pathways regulated in *G. diazotrophicus* when exposed to exudates from sugarcane, revealing regulation of specific responses, as the induction of proteins related to carbohydrate and energy metabolism, and transcription and translation processes (*Lery et al., 2011*; *dos Santos et al., 2010*). However, the molecular mechanisms modulating such interactions are still unclear, particularly those activated during the association with dicot host plants.

The present work aims to evaluate the physiological and molecular responses of *G. diazotrophicus* to cocultivation with *A. thaliana* seedlings. Bacterial growth in response to cocultivation and plant exudates was analyzed. Through comparative proteomic analysis, *G. diazotrophicus* proteins regulated during cocultivation were investigated. The role of some up-accumulated proteins in the response *G. diazotrophicus* to cocultivation was analyzed by reverse genetics, using insertion mutants. The results allowed us to identify mechanisms essential for bacterial response to cocultivation with *Arabidopsis* seedlings.

## MATERIALS AND METHODS

### Plants and growth conditions

Seeds of *A. thaliana* (Col-0) were germinated in vitro in Petri dishes containing half-strength MS medium liquid (*Murashige & Skoog, 1962*) with 0.5% sucrose and 0.05% MES buffer. The seedlings were cultivated for 10 days at 23 °C under 60% relative humidity and 70 µmol m$^{-2}$ s$^{-1}$ light (12 h photoperiod). Roots were not covered from the light during the 12 h photoperiod.

### *G. diazotrophicus* PA15 culture conditions

The wild-type strain of *G. diazotrophicus* PAl5 used in this study comes from the culture collection of the Universidade Estadual do Norte Fluminense Darcy Ribeiro (UENF, Campos dos Goytacazes, Rio de Janeiro State, Brazil). The insertional mutants of *G. diazotrophicus* PAl5, defective in production of the proteins EttA (A9H4G2), AccC (A9HEX0), and Zwf (A9H326) were obtained from the "*G. diazotrophicus* PAl5 mutant library" of the Laboratório de Biotecnologia—UENF (*Intorne et al., 2009*). Such a mutant library was obtained by using EZ-Tn5 <R6Kyori/KAN-2>Tnp insertion kit (Epicentre, Madison, WI, USA).

   *G. diazotrophicus* wild-type and mutant strains were grown in DYGS medium composed of (g. L$^{-1}$) 2.0 glucose; 1.5 bacteriological peptone; 2.0 yeast extract; 0.5 K$_2$ HPO$_4$, 0.5 MgSO$_4$ .7H$_2$O; and 3.75 glutamic acid; with the pH of the medium adjusted to 6.0 (*Reis, Olivares & Döbereiner, 1994*). Bacterial cells were cultivated under constant agitation and temperature (250 min$^{-1}$ and 30 °C) in a shaker (C25 Incubator; New Brunswick Scientific, Edison, NJ, USA) until reaching an optical density (O.D.$_{600}$) equal to 1.0 (~10$^8$ cells. mL$^{-1}$).

### Plant-Bacteria cocultivation

Ten *A. thaliana* seedlings were grown in vitro for 10 days in Petri dishes containing 18 mL of half-strength MS medium plus 0.5% sucrose without the addition of hormones. After reach O.D.$_{600}$ 1.0, *G. diazotrophicus* wild-type and mutant strains cells were washed three times with 0.85% (v/v) NaCl to prevent carryover of spent growth medium. Then, ten 10-day-old *A. thaliana* seedlings were inoculated with *G. diazotrophicus* wild-type and mutant strains at a final concentration of 10$^7$ cells mL$^{-1}$, as described in the previous item (nine repetitions of each treatment, including control). Control treatment was performed by inoculation of *G. diazotrophicus* wild-type and mutant strains at a concentration of 10$^7$ cells mL$^{-1}$ in MS medium without *A. thaliana* seedlings. Petri dishes were maintained at 23 °C under an irradiance of 120 mol/photons m$^{-2}$s$^{-1}$ for 24 h. Bacterial growth was quantified by optical density (O.D.$_{600}$) analysis.

### Proteomic analysis

#### Protein extraction

Three biological replicates of *G. diazotrophicus* cells were collected from cultures in MS medium (control) and from *G. diazotrophicus*/*A. thaliana* cocultivation, and were used for proteomic analysis. For this purpose, 6.5 mL of each cultivation medium sample was centrifuged to collect the bacteria. Protein extraction of all samples was performed as

previously described by *Passamani et al. (2017)*. Specifically, after discarding the supernatant, the pellets were resuspended in 300 µL of the TCA/acetone precipitation buffer (10% trichloroacetic acid in acetone with 20 mM dithiothreitol-DTT) and maintained under constant stirring for 60 min at 4 °C. Samples were, then, maintained at −20 °C for 60 min and centrifuged (30 min; 4 °C; 12.000 g). The supernatants were discarded, and the pellets were rinsed three times with ice-cold acetone containing 20 mM DTT, followed by centrifugation (5 min; 4 °C; 12,000 g). The supernatants were discarded, and the pellets were maintained at room temperature to dry. The pellets were resuspended in 300 µL of urea/thiourea extraction buffer (7M urea, 2M thiourea, 1% DTT, 2% Triton X-100, 5 µM pepstatin, 1 mM phenylmethanesulfonylfluoride-PMSF) and maintained under stirring until complete homogenization. The samples were centrifuged (15 min; 4 °C; 12.000 g), and the supernatants were collected.

The total protein concentration of the samples was estimated using the 2-D Quant Kit (GE Healthcare, Amersham Place, Little Chalfont, UK) following the manufacturer's recommendations. The absorbance of each sample and a standard curve with bovine albumin serum (BSA, GE Healthcare, Amersham Place, Little Chalfont, UK) was determined in a Synergy 2 Multimode Reader (Biotek Instruments, Winooski, VT, USA) at 485 nm. Protein samples were stored in an ultra-freezer at −80 °C.

### Protein digestion

Protein extracts (100 µM) were precipitated in methanol/chloroform as described by *Nanjo et al. (2012)*. Then, protein extracts were digested as previously described by *Passamani et al. (2017)*. Specifically, pellets were resuspended in 25 µL of 0.2% (v/v) RapiGest surfactant (Waters, Milford, CT, USA). Samples were, then, vortexed rapidly and incubated in a heated mixer at 80 °C for 15 min, and 2.5 µL of 100 mM DTT (Bio-Rad Laboratories, Hercules, CA, USA) was added. Subsequently, the samples were vortexed and incubate for 30 min at 60 °C under constant shaking at 350 rpm. Then, 2.5 µL of 300 mM iodoacetamide (GE Healthcare, Piscataway, NJ, USA) was added in each sample. The samples were, then, vortexed rapidly and incubated for 30 min in the dark at room temperature. Subsequently, digestion was performed by adding 20 µL of 50 ng/µL trypsin solution (Promega, Madison, WI, USA) prepared in 50 mM ammonium bicarbonate buffer. The samples were, then, incubated overnight at 37 °C. The precipitation of RapiGest was performed by adding 10 µL of 5% (v/v) trifluoroacetic acid, and the tubes were incubated for 30 min at 37 °C, followed by centrifugation (30 min; 8 °C; 15.000 g). Subsequently, tryptic digestions were desalted using Pierce C18 spin columns (Thermo Scientific, Waltham, MA, USA) according to the manufacturer's instructions. The eluted peptides were then vacuum dried and reconstituted in 50 mM ammonium bicarbonate plus 0.1% formic acid and adjusted to a final concentration of 2 µg µL$^{-1}$. The digested samples were then transferred to total recovery flasks (Waters, Milford, CT, USA).

### LC-MS/MS analyses

LC-MS/MS analyses were performed as previously described by *Passamani et al. (2017)*. Specifically, a liquid nanoACQUITY ultraperformance UPLC connected to a Q-TOF

SYNAPT G2-Si HDMS (Waters, Milford, CT, USA) mass spectrometer was used for LC-MS-HDMS$^E$ analysis. The chromatographic step was performed by injecting two µL of the digested samples for sample normalization before the relative quantification of proteins. In the separation step, the samples (digested protein, two µg) were loaded into the nanoACQUITY UPLC 5 µm C18 (180 µm × 20 mm) column at five µl min$^{-1}$ for 3 min and then into the nanoACQUITY HSS T3 1.8 µm analytical reversed-phase column (75 µm × 150 mm) at 400 nL min$^{-1}$. The temperature of the column was set to 45 °C. For the elution of the peptides; a binary gradient was used: mobile phase A consisted of water (Tedia, Fairfield, OH, USA) and 0.1% formic acid (Sigma-Aldrich, St. Louis, MO, USA), and mobile phase B consisted of acetonitrile (Sigma-Aldrich, St. Louis, MO, USA) and 0.1% formic acid. Gradient elution was performed as follows: 7% B over 3 min, increased from 7 to 40% B at 90.09 min; then 40 to 85% B at 94.09 min; held constant at 85% until 98.09 min; decreased to 7% B at 100.09 min, and finally held steady at 7% B until the end of the run at 108.09 min. The mass spectrometer was operated in resolution mode (V mode) and positive mode with ionic mobility; with collision energy transfer from 19 to 45 V in high-energy mode; a voltage and capillary cone of 30 V and 2,800 V, respectively; and a source temperature of 70 °C. In the TOF parameters, the scan time was set to 0.5 s in continuous mode, with a mass range of 50 to 2,000 Da. Human [Glu1]-fibrinopeptide B (Sigma-Aldrich , St. Louis, MO, USA) at 100 fmol.µL$^{-1}$ was used as an external calibrant, with acquisition performed every 30 s. Mass spectrum acquisition was performed by using MassLynx v4.0 software.

### Proteomic data analysis

Spectral processing and database searching were made as previously described by *Passamani et al. (2018)*, through ProteinLynx Global Server (PLGS; version 3.0.2) (Waters, Milford, CT, USA) and ISOQuant workflow software (*Distler et al., 2014, 2016*). Specifically, the PLGS analysis was performed with the follows processing parameters: a low energy threshold of 150 (counts), a high energy threshold of 50, and an intensity threshold of 750. Additionally, the analysis was performed with the following parameters adjustment were made: two missed cleavages, minimum fragment ions per peptide equal to 3, minimum ions per peptide equal to 7, minimum peptides per protein equal to 2, fixed carbamidomethyl (C) modifications and variable modifications of oxidation (M) and phosphorylation (STY). The false discovery rate for the identification of peptides and proteins was established according to a maximum of 1%, with a minimum peptide length of six amino acids. Proteomic data were processed using the *G. diazotrophicus* RIOGENE proteome database (www.uniprot.org/proteomes/UP000001176).

After data analysis in ISOQuant, only the proteins that were present or absent (for single proteins) in all three biological replicates were considered in the differential abundance analysis. Data were analyzed using Student's *t*-test (two-tailed) as previously described by *Leandro et al. (2019)*. Specifically, proteins with *p*-values < 0.05 were considered to be up-accumulated if the fold change (FC) was higher than 1.5 and down-accumulated if the FC was less than 0.667. Protein network analyses was performed using STRING database

with confidence as the meaning of network edges and 0.700 of interaction score (high confidence) (www.string-db.org).

## Sample preparation of the plant, bacterial and cocultivation exudates

Exudates from *G. diazotrophicus* and *A. thaliana cultivations*, and *G. diazotrophicus/A. thaliana* cocultivations were obtained, as illustrated in Fig. S1. Three biological replicates of each treatment were performed.

Arabidopsis *thaliana* seedlings were cultivated in Petri dishes containing 18 mL of half-strength MS medium at 23 °C for 10 days. Plant exudates were collected by filtration of the cultivation medium with syringe filters (0.22 μm) and stored at −80 °C.

In order to obtain *G. diazotrophicus* exudates, the bacteria were inoculated in Petri dishes containing 18 mL of half-strength MS medium and cultivated for 24 h at 23 °C. When the bacterial cultures reached an O.D.$_{600nm}$ = 1.0, the culture was transferred to centrifuge tubes and centrifuged at 8,000×$g$ for 5 min. The supernatant was filtered with a syringe filter (0.22 μm) and stored at −80 °C.

Exudates from the cocultivation of *A. thaliana* seedlings and *G. diazotrophicus* were obtained from cocultivation assays performed as previously described in item 2.3. After cocultivation, the medium was collected and centrifuged at 8,000×$g$ for 5 min. The supernatant was filtered with a syringe filter (0.22 μm) and stored at −80 °C.

## Analysis of the *G. diazotrophicus* PA15 response to exudates

Exudates from *G. diazotrophicus*, *A. thaliana* seedlings, or cocultivation, were previously obtained as described in "Sample Preparation of the Plant, Bacterial and Cocultivation Exudates." Such exudates were added to *G. diazotrophicus* cultivation, as shown in Fig. S1. *G. diazotrophicus* culture (O.D.$_{600nm}$ = 1.0) produced as described in "G. diazotrophicus PA15 Culture Conditions" was inoculated in 50 mL Erlenmeyer flasks containing half-strength fresh MS medium and each exudate, as the following final proportion: 10% (*G. diazotrophicus* culture), 45% (fresh MS medium) and 45% (exudate). Control samples received 10% of *G. diazotrophicus* and 90% of half-strength fresh MS medium. All cultures had a final volume of 10 mL. The cultures were grown at 30 °C, with constant agitation (250 rpm) in an orbital shaker. Bacterial growth was analyzed by optical spectrometry (O.D.$_{600}$). Three biological replicates were performed for each treatment.

## Phylogenetic analyses and genomic organization of accC and ettA

The sequences of AccC and EttA proteins coding genes of *G. diazotrophicus* (RefSeq: NC_010125.1), and related bacterial species were obtained through the NCBI database. The *accC* and *ettA* locus of *G. xylinus* NBRC 3288 (RefSeq: NC_016037), *Azospirillium brasilense* Sp245 (RefSeq: NZ_CP022253.1), *Herbaspirillum seropedicae* SmR1 (RefSeq: NC_014323), *Pseudomonas syringae* pv. tomato str. DC3000 (RefSeq: NC_004578.1), and *Escherichia coli* ATCC8739 (RefSeq: NZ_CP022959.1) were used for the analysis of phylogeny, gene context and operon prediction.

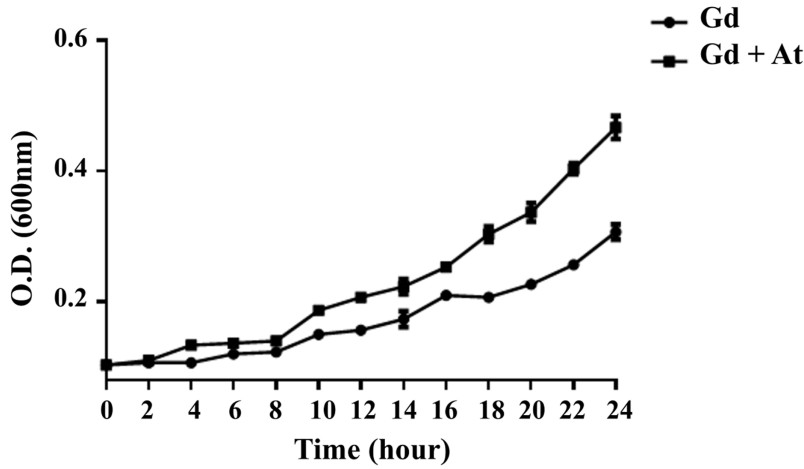

**Figure 1** ***G. diazotrophicus* growth increases in response to cocultivation with *A. thaliana* seedlings.** *G. diazotrophicus* (Gd) was co-cultivated with *A. thaliana seedlings* (At), and its growth was analyzed every 2 h during the first 24 h of cocultivation.

Gene sequences were aligned with MEGA-X software version 10.1.7 (*Kumar et al., 2018*) using the ClustalW algorithm (default settings). MEGA-X was also used to select the best substitution models for phylogenetic analyses, and to generate a phylogenetic tree using the maximum likelihood (ML) method. The parameters selected for *accC* and *ettA* phylogenetic threes were the models of substitution gama time reversible (GTR) with the invariant site (I) and GTR with the gamma-distributed site, respectively, with 1,000 bootstraps.

The KEGG (https://www.genome.jp/kegg/) and Microbes databases (http://www.microbesonline.org/) were used for an in silico prediction of the operon organization.

## Statistical analyses

The assays were performed with nine replicates in each condition. The growth rate data of *G. diazotrophicus* wild-type and mutant strains under the control and cocultivation treatments were subjected to a mean test (Tukey) at the 5% probability level to measure significance between treatments. Data analyses were performed by using GraphPad Prism v. 7.00.159 (GraphPad Software, La Jolla, CA, USA).

## RESULTS

### *G. diazotrophicus* growth increases in response to cocultivation with *A. thaliana*

*G. diazotrophicus* was co-cultivated with *A. thaliana* seedlings, and its growth was analyzed during the first 24 h of the association. Figure 1 shows that the presence of *A. thaliana* seedlings increased bacterial growth by approximately 45% after 24 h of cocultivation. These data suggest that plant-produced metabolites are beneficial to bacterial growth. Microscopy analyses did not show morphological differences in *G. diazotrophicus* in response to cocultivation with *A. thaliana* (Fig. S2).
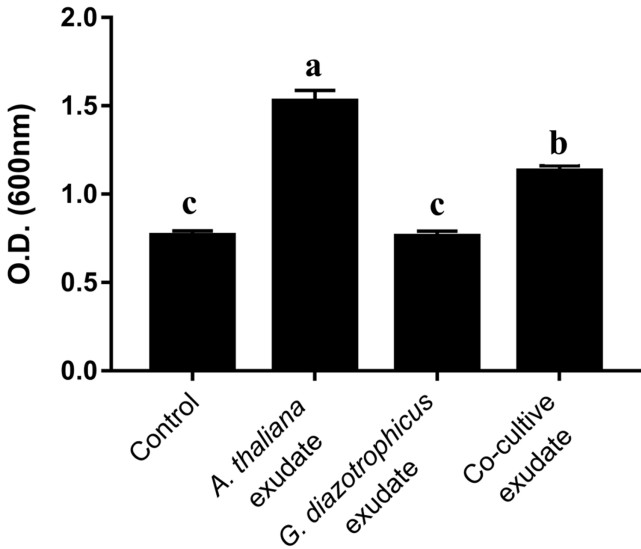

**Figure 2 *A. thaliana* exudates that increase *G. diazotrophicus* growth are produced constitutively.**
Exudates from non-inoculated *A. thaliana* seedlings, from *G. diazotrophicus* cultures, and from *G. diazotrophicus/A. thaliana* cocultivation were added to new *G. diazotrophicus* cultures and its growth was analyzed after 24 h of treatment. Control treatment represents bacterial growth in the absence of exudates. Different letters indicate significant differences from the control by the Tukey test at 5% probability ($n = 9$).

### *A. thaliana* seedlings constitutively exude metabolites that increase *G. diazotrophicus* growth

To verify whether the exudation of beneficial metabolites by *A. thaliana* seedlings is constitutive or is induced by the presence of *G. diazotrophicus*, exudates from *G. diazotrophicus/A. thaliana* cocultivation were compared to those produced by non-inoculated seedlings and by *G. diazotrophicus* cultured alone. Thus, MS media containing exudates from non-inoculated seedlings, from *G. diazotrophicus* cultures, and from *G. diazotrophicus/A. thaliana* cocultivation were obtained and added to new *G. diazotrophicus* cultures as shown in Fig. S1. As a control, *G. diazotrophicus* was grown in the absence of exudates. Figure 2 shows that exudates from non-inoculated *A. thaliana* (*A. thaliana* exudate) and *G. diazotrophicus/A. thaliana* cocultivation (Co-cultive exudate) resulted in a similar increase in bacterial growth. As expected, *G. diazotrophicus* cultured alone (*G. diazotrophicus* exudate) did not improve bacterial growth. These results revealed that *A. thaliana* seedlings constitutively exude metabolites that are beneficial for bacterial growth.

### The proteomic profile of *G. diazotrophicus* alters in response to cocultivation with *A. thaliana*

The main *G. diazotrophicus* proteins regulated during cocultivation with *A. thaliana* seedlings were investigated by comparative proteomic analysis. For this purpose, total protein extracts obtained from co-cultivated bacteria were compared to those of bacteria cultivated in the absence of plants. A total of 450 bacterial proteins were identified

**Table 1 G. diazotrophicus proteins regulated in response to cocutiltivation with A. thaliana.**

| Accession | Description | Gene | Reported peptides | Max score | Fold change |
|---|---|---|---|---|---|
| Up-accumulated | | | | | |
| A9HSA5 | Putative thioredoxin protein | GDI3107 | 4.00 | 3,216.44 | 1.50 |
| RL23 | 50S ribosomal protein L23 | rplW | 6.00 | 8,093.88 | 1.52 |
| A9HIP1 | S-(hydroxymethyl)glutathione dehydrogenase | frmA | 4.00 | 1,057.50 | 1.52 |
| A9H0G0 | Glucose-6-phosphate 1-dehydrogenase | zwf | 27.00 | 5,629.12 | 1.54 |
| RS2 | 30S ribosomal protein S2 | rpsB | 16.00 | 11,963.12 | 1.56 |
| A9HDU1 | Oxidoreductase domain protein | GDI1200 | 14.00 | 6,904.23 | 1.63 |
| A9HAZ8 | Protein TolR | tolR | 3.00 | 1,454.51 | 1.64 |
| A9HJY0 | Putative transcriptional Regulator, MarR family | GDI_2027 | 4.00 | 6,080.51 | 1.67 |
| A9H0W3 | Adenylyl-sulfate kinase | cysC | 21.00 | 4,170.78 | 1.68 |
| A9HM86 | Glycine–tRNA ligase beta subunit | glyrs | 11.00 | 1,189.10 | 1.69 |
| A9HS68 | Signal recognition particle protein | ffh | 12.00 | 2,042.15 | 1.75 |
| A9HGY2 | Putative Squalene–hopene cyclase | GDI1620 | 2.00 | 350.53 | 1.77 |
| A9HJS1 | Uncharacterized protein | GDI1999 | 4.00 | 769.19 | 1.77 |
| RS8 | 30S ribosomal protein S8 | rpsH | 6.00 | 4,884.06 | 1.81 |
| SYH | Histidine–tRNA ligase | hisS | 6.00 | 728.77 | 1.82 |
| A9H324 | 6-phosphogluconate dehydrogenase, decarboxylating | gnd | 16.00 | 20,323.17 | 1.87 |
| ASSY | Argininosuccinate synthase | argG | 10.00 | 4,334.52 | 1.87 |
| A9HRE6 | Putative metallopeptidase | GDI2948 | 19.00 | 3,042.59 | 1.98 |
| RL6 | 50S ribosomal protein L6 | rplF | 9.00 | 8,968.96 | 2.03 |
| A9H397 | 2,3-bisphosphoglycerate-independent phosphoglycerate mutase | gpmI | 11.00 | 4,303.62 | 2.03 |
| A9HJB6 | Dihydrolipoyl dehydrogenase | lpdA | 8.00 | 1,282.86 | 2.09 |
| A9HM48 | Glycine dehydrogenase (decarboxylating) | gcvP | 20.00 | 1,964.67 | 2.14 |
| A9H7Z5 | Glutamine synthetase | glnA | 23.00 | 15,678.34 | 2.20 |
| A9HII0 | Orotate phosphoribosyltransferase | pyrE | 5.00 | 3,799.75 | 2.29 |
| A9H108 | Glutamate–cysteine ligase | GDI3250 | 7.00 | 1,745.48 | 2.30 |
| A9H459 | 30S ribosomal protein S1 | rpsA | 30.00 | 11,955.40 | 2.38 |
| RL14 | 50S ribosomal protein L14 | rplN | 4.00 | 2,991.02 | 2.38 |
| A9H326 | Glucose-6-phosphate 1-dehydrogenase | zwf | 5.00 | 837.57 | 2.53 |
| A9H3M8 | 50S ribosomal protein L5 | rplE | 4.00 | 1,646.31 | 2.58 |
| CLPX | ATP-dependent Clp protease ATP-binding subunit ClpX | clpX | 11.00 | 3,303.25 | 2.64 |
| ISPG | 4-hydroxy-3-methylbut-2-en-1-yl diphosphate synthase (flavodoxin) | ispG | 6.00 | 1,571.93 | 3.00 |
| A9GZU8 | Conserved protein | GDI0061 | 7.00 | 1,962.62 | 3.04 |
| A9HEX0 | acetyl-CoA carboxylase biotin carboxylase subunit | accC | 13.00 | 4,423.06 | 3.52 |
| A9HS02 | Elongation factor G | fusA | 31.00 | 10,183.21 | 3.93 |
| A9H932 | TonB-dependent receptor | GDI0667 | 14.00 | 2,074.37 | 4.10 |
| ILVC | Ketol-acid reductoisomerase (NADP(+)) | ilvC | 5.00 | 2,463.85 | 4.21 |
| A9H4G2 | Energy-dependent translational throttle A protein EttA | ettA | 5.00 | 506.79 | 4.23 |
| A9HEZ2 | Aconitate hydratase | acnA | 44.00 | 10,076.64 | 8.49 |
| A9HNX4 | 5-methyltetrahydropteroyltriglutamate–homocysteine methyltransferase | metE | 38.00 | 11,719.34 | 9.56 |
| Down-accumulated | | | | | |

(Continued)

| Table 1 (continued) | | | | | |
| --- | --- | --- | --- | --- | --- |
| Accession | Description | Gene | Reported peptides | Max score | Fold change |
| A9HB99 | Uncharacterized protein | GDI0843 | 4.00 | 10,074.54 | 0.44 |
| A9HPF6 | Porin | oprB | 9.00 | 2,838.53 | 0.46 |
| A9H577 | Sugar ABC transporter substrate-binding protein | GDI0354 | 8.00 | 4,390.16 | 0.47 |
| A9H073 | Alcohol dehydrogenase GroES domain protein | GDI_3142 | 4.00 | 1,094.65 | 0.48 |
| A9HRF1 | Succinate–CoA ligase (ADP-forming) subunit alpha | sucD | 5.00 | 2,866.13 | 0.51 |
| A9HPH9 | 10 kDa chaperonin | groS | 5.00 | 4,835.59 | 0.54 |
| A9HNP0 | D-xylose ABC transporter, periplasmic substrate-binding protein | xylF | 18.00 | 13,607.54 | 0.58 |
| A9HK34 | Cold-shock DNA-binding domain protein | GDI2048 | 5.00 | 34,042.72 | 0.59 |
| A9HEI5 | Uncharacterized protein | GDI1295 | 3.00 | 3,273.24 | 0.63 |
| A9H9C0 | Inosine-guanosine kinase | GDI0702 | 4.00 | 1,497.69 | 0.63 |
| A9HL73 | Alanine–tRNA ligase | alaS | 13.00 | 1,109.99 | 0.64 |
| A9HPE1 | Extracellular solute-binding protein family 1 | GDI2634 | 13.00 | 9,768.71 | 0.66 |

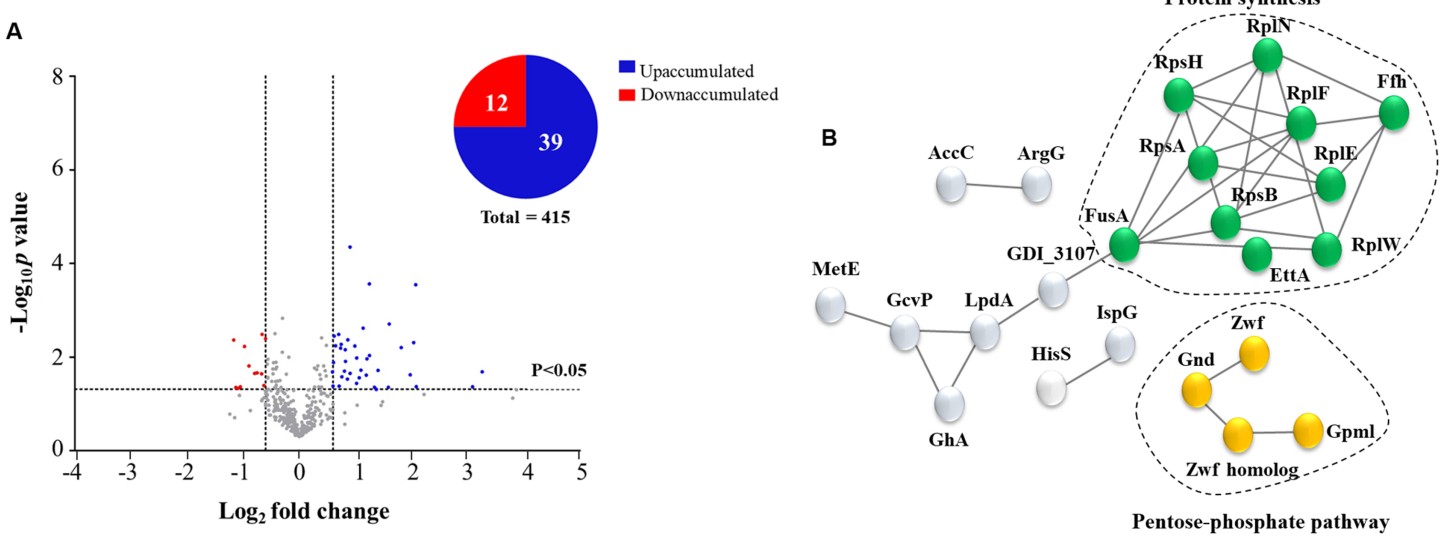

**Figure 3 Analyses of identified proteins of *G. diazotrophicus* co-cultivated with *A. thaliana* seedlings.** Volcano plot of all identified proteins and graphical representation of the percentages of DAPs up-accumulated (blue) and down-accumulated (red) of *G. diazotrophicus* co-cultivated with *A. thaliana* (A). The spots represent differential abundance (log2 fold change) of identified proteins in function of statistical significance (−log10 *p*-value). (B) Protein networks of DAPs up-accumulated during cocultivation. Colored proteins represent clustered proteins from "protein synthesis pathway" (green) and "pentose-phosphate pathway" (yellow).

(Table S1). Among these proteins, 51 differentially accumulated proteins (DAPs) were observed, with 39 proteins increased, and 12 proteins decreased (Table 1; Fig. 3A).

## Protein networks regulated during cocultivation

The functional association networks of the regulated proteins were analyzed with the software String (version 10.5). The analyses were performed for both the up-accumulated and down-accumulated proteins using a confidence level of 0.7. Up-accumulated protein

analysis revealed two major protein groups: "pentose-phosphate pathways" and "ribosomal proteins" (Fig. 3B). When similar analyses were performed for down-accumulated proteins, no interaction networks between the proteins were identified (Fig. S3).

## Main up-accumulated proteins

Proteins that are potentially relevant to plant-bacteria interactions were up-accumulated during the first 24 h of cocultivation. Among these proteins, four showed the greatest up-regulation: 5-methyltetrahydropteroyltriglutamate-homocysteine methyltransferase (MetE), which is involved in the synthesis of methionine; aconitate hydratase (AcnA), which is associated with the metabolism of tricarboxylic acids; energy-dependent translational throttle A (EttA), which is involved in the regulation of ribosomal activity, and may also be necessary for protection against antimicrobial compounds; ketol-acid reductoisomerase (IlvC), which is involved in amino acid metabolism (Table 1).

Other up-accumulated proteins included Zwf (glucose-6-phosphate 1-dehydrogenase) and Gnd (6-phosphogluconate dehydrogenase decarboxylating), which are related to the pentose-phosphate pathway (Table 1). Another protein that was increased, ClpX (ATP-binding subunit of the ClpXP protease), is involved in the degradation of unfolded proteins and is potentially related to bacterial quorum sensing (Table 1). Additionally, a key protein for de novo fatty acid biosynthesis, AccC (acetyl-CoA carboxylase biotin carboxylase subunit), and a protein involved with nutrients uptake (TonB-dependent receptor) were also up-accumulated (Table 1).

## Down-accumulated proteins

Only 12 proteins were down-accumulated in response to cocultivation. Among these proteins, we highlight three transporters related to the import of sugars: OprB, a Sugar ABC transporter substrate-binding protein, and the D-xylose ABC transporter substrate-binding protein (Table 1).

## Interruption of ettA and accC impairs *G. diazotrophicus* growth-response to cocultivation with *A. thaliana*

Using reverse genetic analysis by mutagenesis approach, we checked whether specific proteins, up-accumulated in our proteomic analysis, are essential for *G. diazotrophicus* response to cocultivation with *A. thaliana*. Mutant strains defective for proteins potentially relevant for bacterial growth/multiplication were selected. *G. diazotrophicus* mutant strains defectives for EttA (Δ*ettA*), AccC (Δ*accC*), and Zwf (Δ*zwf*) were co-cultivated with *A. thaliana* seedlings, and their growth was analyzed in comparison to wild-type. Figure 4 shows that the lack of Zwf did not affect *G. diazotrophicus* growth induction in response to cocultivation with *A. thaliana*. On the other hand, the lack of EttA and AccC implies the loss of growth induction response, suggesting a key role of these proteins in this process. Additionally, the growth of Δ*accC* in the control treatment (MS) was significantly lower than the wild-type strain, suggesting that AccC is also important to bacterial growth under normal culture conditions.

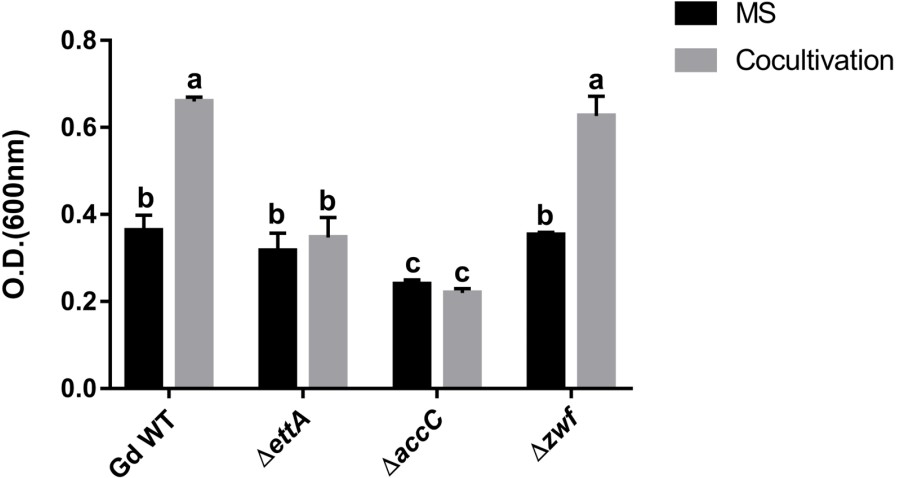

**Figure 4 Reverse genetics revealed *G. diazotrophicus* genes involved with response to *A. thaliana* cocultivation.** Insertional mutants of *G. diazotrophicus* defectives in production of three proteins up-accumulated in our proteomic analysis were selected to perform cocultivation assay with *A. thaliana* seedlings. The growth of each strain was analyzed after 24 h of cocultivation. Different letters indicate significant differences from the control by the Tukey test at 5% probability (*n* = 9).

## In silico prediction suggests that ettA and accC are organized in operons in *G. diazotrophicus* genome

Phylogenetic analysis results for both *ettA* and *accC* genes showed the formation of two distinct groups (Fig. 5). The sequences of *G. diazotrophicus* are closer related to *G. xylinus* and *A. brasiliense*. Although *H. seropedicae* is a PGPB as *G. diazotrophicus*, its gene sequences are closer to those from the phytopathogen *P. syringae* and the human pathogen *E. coli* (Fig. 5). Moreover, operon prediction results suggest that both *ettA* and *accC* are organized in operons in *G. diazotrophicus* genome. The operon that contains *accC* is composed of three genes (*aroQ*, *accB*, *accC*) in almost all the bacterial species analyzed, except *E. coli*, which presents only two genes in this operon (*accB* and *accC*) (Fig. 5A). In all bacterial species analyzed, *accC* is in the last position of the operon (Fig. 5A). So, the interruption in *accC* probably does not compromise the transcription of the other components of such operon.

In almost all the bacterial species analyzed, *ettA* is not located in an operon, except in *G. diazotrophicus* and *G. xylinus* (Fig. 5B). In these two species, *ettA* is located in an operon, upstream of a gene that codifies a GCN5-related N-acetyltransferase (GNAT) family protein (Fig. 5B). So, the interruption of *ettA* may compromise the production of a GNAT family protein in *G. diazotrophicus*.

## DISCUSSION

The present work aimed to evaluate the response of *G. diazotrophicus* to cocultivation with *A. thaliana* seedlings with an emphasis on the main bacterial proteins regulated by the interaction. Our results revealed the induction of bacterial growth during cocultivation, suggesting a beneficial effect of metabolites exuded by the plants. The data also revealed

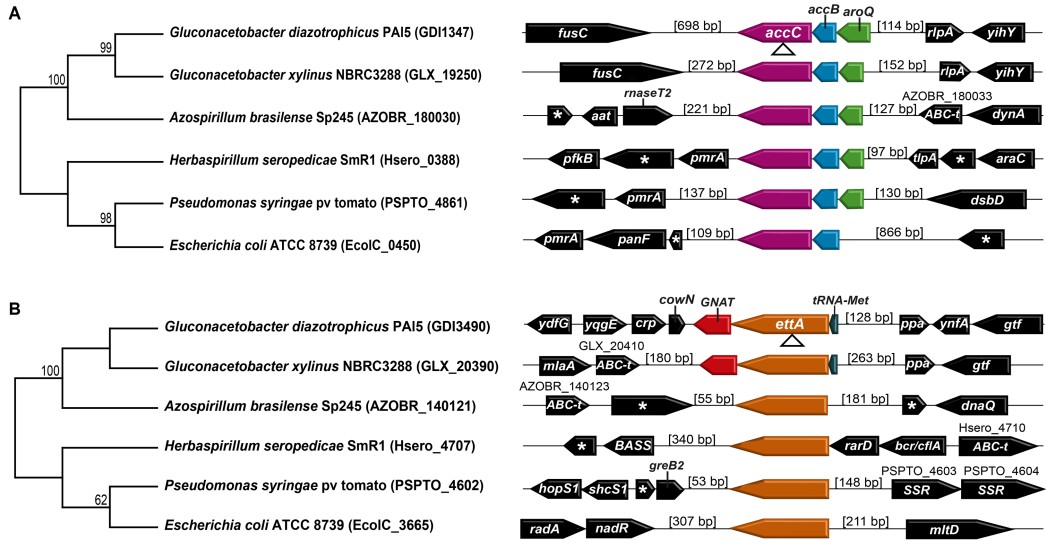

**Figure 5 Phylogenetic analysis and in silico operon prediction of *accC* and *ettA*.** Phylogenetic relationship of the nucleotide sequences and gene cluster flanking *accC* (A) and *ettA* (B) in *G. diazotrophicus* PAl 5 (RefSeq: NC_010125.1) and related bacterial species following the RefSeq NCBI database: *G. xylinus* NBRC 3288 (RefSeq: NC_016037), *A. brasilense* Sp245 (RefSeq: NZ_CP022253.1), *Herbaspirillum seropedicae* SmR1 (RefSeq: NC_014323), *P. syringae* pv. tomato str. DC3000 (RefSeq: NC_004578.1), and *E. coli* ATCC 8739 (RefSeq: NZ_CP022959.1). The numbers at the nodes indicate bootstrap support. Arrows indicate the orientations of genes translation. Codes above genes are the locus tag of the gene sequences that do not have a described abbreviation. Asterisks indicate uncharacterized genes. *Triangles*: transposon insertion sites; *SSR*, Site-specific recombinases, phase integrase family genes; *ABC-t*, ABC-transporter family genes; *GNAT*, GCN5-related N-acetyltransferases family genes; *BASS*, Bile Acid: Na+ Symporter family gene.

that the production of such plant-beneficial exudates is a constitutive process. Comparative proteomic analyses demonstrated the regulation of bacterial protein pathways that were mainly related to sugar metabolism (pentoses) and protein synthesis. Among the regulated proteins, MetE, AcnA, IlvC, EttA, Zwf, ClpX, and AccC deserve special attention.

Cocultivation with *A. thaliana* seedlings resulted in increased growth of *G. diazotrophicus* (Fig. 1). Studies that support this finding have been carried out with *G. diazotrophicus* under cocultivation with sugarcane plants, where the induction of bacterial multiplication has been reported (*Lery et al., 2011*; *dos Santos et al., 2010*). The substances exuded by plant roots include sugars, amino acids, mucilage, flavonoids, organic acids, and volatile compounds capable of attracting bacteria and favoring their multiplication (*Sasse, Martinoia & Northen, 2018*). Important functions have been attributed to exuded sugars, which would serve as a source of energy for microorganisms (*Dennis, Miller & Hirsch, 2010*; *Sasse, Martinoia & Northen, 2018*; *Badri & Vivanco, 2009*). However, a sugar-rich culture medium (0.5% sucrose) was used in the present work. Nevertheless, the presence of exudates from *A. thaliana* seedlings positively modulated the growth of *G. diazotrophicus*, suggesting contributions of other beneficial metabolites to this process.

The analysis of *G. diazotrophicus* growth in the presence of exudates from plants and from plant-bacteria co-cultivation demonstrated that the production of beneficial metabolites by plants is constitutive (Fig. 2). *Jacoby, Martyn & Kopriva (2018)* showed that *A. thaliana* roots secrete a large number of secondary metabolites that are used by bacteria. The amino acids valine, tryptophan, threonine, and glutamine, in addition to pyrimidine and purine derivatives including adenosine, cytidine, guanine, and uridine, are consumed by plant-associated bacteria (*Jacoby, Martyn & Kopriva, 2018*).

Proteomic analyses revealed changes in the protein profile of *G. diazotrophicus* in response to cocultivation, with 39 proteins classified as up-accumulated and 12 as down-accumulated. Among the most up-accumulated proteins was 5-methyltetrahydropteroyltriglutamate-homocysteine methyltransferase (MetE), which is involved in the conversion of homocysteine to methionine (*Winzer, 2002*). Additionally, MetE is related to the production of AI-2 quorum sensing (QS) molecule in different microorganisms (*Winzer, 2002*). However, similar to the rest of the Alphaproteobacteria, *G. diazotrophicus* lacks a LuxS homolog required for AI-2 synthesis (*Rezzonico & Duffy, 2008*), so the up-accumulation of MetE in our proteomic analysis should not be related to QS activity. Another up-accumulated protein was Aconitate hydratase (AcnA), which is associated with the metabolism of tricarboxylic acids and may protect bacterial cells from reactive oxygen species (*Doi & Takaya, 2015*). Ketol-acid reductoisomerase (IlvC), which was also up-accumulated, plays an essential role in the pathway of amino acid biosynthesis (*Li et al., 2017*). These three proteins were also detected among the most up-accumulated proteins of *G. diazotrophicus* during its cocultivation with sugarcane plants (*Lery et al., 2011*). Our results highlight the MetE, IlvC, and AcnA proteins as potentially relevant for the association of *G. diazotrophicus* with both monocot and dicot plants.

Approximately 23% of the up-accumulated proteins are ribosomal (RpsA, RpsB, RpsH, RplE, RplF, RplN, and RplW), indicating a positive effect of cocultivation on bacterial protein synthesis. In *Escherichia coli*, the genes coding for RpsH, RplE, RplF and RplN are in an operon called *spc* that codes for 11 ribosomal proteins, essentials to protein synthesis machinery (*Aseev & Boni, 2011*). RpsA interacts with the mRNA leader sequence during the formation of the translation initiation complex, acting directly in the regulation of such a process (*Komarova et al., 2002*). RpsA is also positively regulated in the endophytic bacterium *H. seropedicae* when exposed to extracts from sugarcane (*Cordeiro et al., 2013*). RplW is essential for bacterial multiplication and is associated with the "Trigger Factor" in ribosomes (*Kramer et al., 2002*; *Tischendorf, Zeichhardt & Stöffler, 1974*). Our data, therefore, suggest the activation of protein synthesis as a critical response to cocultivation, supporting the increased bacterial growth observed.

The proteomic analysis also revealed the up-accumulation of the ABC-F EttA protein that modulates bacterial ribosomes activity in an energy-dependent manner (*Boël et al., 2014*). In *Escherichia coli* cells, the accumulation of EttA increases during the stationary phase, when energetic resources decline (*Boël et al., 2014*). In such a condition, EttA attaches to the ribosome and inhibits protein synthesis (*Boël et al., 2014*). In contrast, the increase in energy availability leads to the dissociation of EttA of the ribosome, allowing it to enter the elongation cycle resuming protein synthesis (*Boël et al., 2014*). Interestingly,

even with our results showing that after 24 h of cultivation *G. diazotrophicus* still has available energy resources, characterized by its exponential growth phase, our proteomic analysis showed the up-accumulation of EttA. This result suggests that in *G. diazotrophicus* the regulation of EttA can be independent of energy availability.

Additionally, the reverse genetics results showed that Δ*ettA* mutant loses the growth response to cocultivation with *A. thaliana*. *ettA* gene seems to belong to an operon in *G. diazotrophicus* genome, located upstream of a gene that codifies a GCN5-related N-acetyltransferase (GNAT) family protein. The phenotype observed for the Δ*ettA* mutant may be the sum of effects caused by the absence of these two proteins (EttA and GNAT). GNATs are widespread in eukaryotes and prokaryotes and are associated with many bacterial processes, such as drug resistance, stress reaction, and regulation of transcription (*Xie et al., 2014*; *Favrot, Blanchard & Vergnolle, 2016*). The result obtained to Δ*ettA* allow us to speculate about the role of EttA in the perception of plant metabolites and the subsequent activation of protein synthesis. However, further analyses are necessary to investigate the participation of the protein GNAT in such a process.

Proteins of the pentose-phosphate pathway were also up-accumulated. This pathway is essential for cell metabolism, including the maintenance of carbon homeostasis and the provision of precursors for nucleotide and amino acid biosynthesis (*Pickl & Schönheit, 2015*; *Stincone et al., 2015*). According to our data, four members of this pathway were up-accumulated: Zwf (Glucose-6-phosphate 1-dehydrogenase), Gnd (6-phosphogluconate dehydrogenase, decarboxylating), Gpml (2,3-bisphosphoglycerate-independent phosphoglycerate mutase) and a Zwf homolog (A9H0G0). Among these proteins, Zwf and Gnd function in the oxidative phase of the pentose-phosphate pathway, resulting in the production of NADPH (*Lim et al., 2002*). Despite the positive regulation of this pathway, our reverse genetics results demonstrate that the induction of *G. diazotrophicus* growth in response to *A. thaliana* cocultivation was maintained even with the lack of Zwf (A9H326). We hypothesize that the activity of the Zwf homolog (A9H0G0), also up-accumulated in our proteomic analysis, may compensate the absence of A9H326 in the pentose-phosphate pathway.

Additionally, among the proteins up-accumulated in our proteomic analysis potentially involved with the bacterial response to plant exudates, ClpX, and AccC deserve attention. ClpX plays a role in the unfolding and degradation of other proteins. Additionally, it has been shown in *Burkholderia cenocepacia* that a mutation in the ClpX coding sequence produces an increase in the production of acyl homoserine lactone, an important group of signaling molecules for quorum sensing in Gram-negative bacteria (*Veselova et al., 2016*). Proteins responsible for quorum sensing are regulated in the presence of hosts and participate in the control of bacterial growth and the regulation of virulence mechanisms (*Sasse, Martinoia & Northen, 2018*; *Pérez-Montaño et al., 2013*; *Steindler et al., 2009*). Additional analyses are necessary to investigate the role of ClpX in controlling unfolding proteins or in quorum sensing mechanisms in the responses of *G. diazotrophicus* to metabolites exuded from *A. thaliana* seedlings.

AccC is a component of bacterial acetyl coenzyme-A carboxylase (ACCase) that catalyzes the first step in fatty acid biosynthesis (*Cheng et al., 2009*). AccC participates in

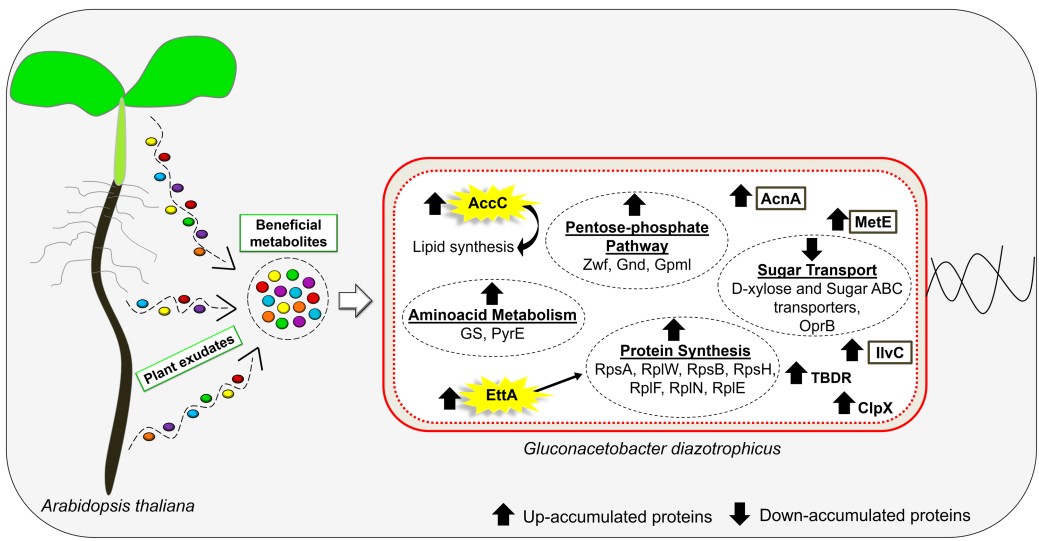

**Figure 6 Schematic illustration of the main responses of *G. diazotrophicus* to beneficial metabolites exuded by *A. thaliana*.** Colored circles represent metabolites exuded from the plant. Brown boxes indicate *G. diazotrophicus* proteins that have also been up-accumulated in proteomic studies performed with other plant species.

bacterial lipid biosynthesis and consequently impacts cellular membrane formation, essential processes for cell viability and bacterial growth (*Cheng et al., 2009*). Once *G. diazotrophicus* growth increased in response to *A. thaliana* cocultivation, the increase in accumulation of AccC observed in our proteomic analyses was expected. Our data using a defective mutant for AccC (Δ*accC*) revealed a lower growth performance in comparison to the wild-type strain. Bacterial multiplication requires the synthesis of lipids to supply the formation of new cell membranes, which may explain our results. Interestingly, another *accC* homolog is present in the genome of *G. diazotrophicus* (GDI0201). However, the corresponding protein was not detected in our proteomic analyzes. The presence of such a homolog gene in the genome of *G. diazotrophicus* did not compensate for the phenotype of Δ*accC* in response to *A. thaliana* cocultivation observed in our results, suggesting a distinct functional role.

Moreover, four proteins related to nutrients uptake were regulated in our proteomic analysis. Among these, the protein TonB-dependent receptor (TBDRs) was highly up-accumulated. TBDRs were described in early studies as an iron transporter. However, currently, its activity is also associated with the active import of other nutrients, such as vitamins B1 and B12 (*Schauer, Rodionov & De Reuse, 2008*). *Molina et al. (2005)*, in a study with corn, showed that the efficiency of initial steps of seed and root colonization by *P. putida* is dependent of TBDR activity, once essential mechanisms to this process, as biofilm formation, are highly iron-dependent. Additionally, the up-accumulation of TBDRs was previously reported in *G. diazotrophicus* in response to sugarcane cocultivation (*Lery et al., 2011*). Thus, our result reinforces the importance of TBDRs in plant-bacteria associations.

The presence of sugars and other carbon structures in root exudates has been described as the main factor in the induction of bacterial multiplication during the initial steps of

plant-bacteria interaction (*Babalola, 2010*). Moreover, in our assays, a high concentration of sugar was provided in the culture medium. Our proteomic analysis showed that proteins related to sugars uptake (OprB, Sugar ABC transporter substrate-binding, and D-xylose ABC transporter substrate-binding) were down-accumulated in *G. diazotrophicus* cells co-cultivated with *A. thaliana*. The up-accumulation of pentose-phosphate pathway proteins in our proteomic analysis reinforces that *G. diazotrophicus* cells were under high availability of carbon sources in our assays. Previously studies demonstrate that bacterial cells growing under high availability of carbon source exhibit high rates of carbon uptake and metabolism (*Russell & Cook, 1995*). In this sense, further analyzes are necessary to understand the role of the down-accumulation of proteins involved with sugar uptake at the beginning of the association between *G. diazotrophicus* and *A. thaliana* seedlings.

## CONCLUSIONS

Taken together, our data provide physiological and molecular aspects related to the association between *G. diazotrophicus* and *A. thaliana* seedlings. As summarized in Fig. 6, *A. thaliana* seedlings constitutively exude beneficial metabolites that increase bacterial growth. These data reinforce the potential involvement of the MetE, AcnA, IlvC, and TonB-dependent receptor proteins as a conserved mechanism during the interaction of *G. diazotrophicus* with both monocot and dicot plants. Additionally, our results revealed the essential role of the proteins EttA and AccC in the bacterial growth activated in response to cocultivation with *A. thaliana*. This is the first study to demonstrate the participation of EttA and AccC proteins in plant-bacteria interactions. Our results open new perspectives for further investigations about the initial steps of such associations in other bacteria and plant species.

### Funding

This work was supported by the Coordination for the Improvement of Higher Education Personnel (CAPES); the Brazilian National Council for Scientific and Technological Development (CNPq); the Rio de Janeiro Research Foundation (FAPERJ); the Funding Authority for Research and Projects (FINEP); and the State University of North Fluminense "Darcy Ribeiro" (UENF) for granting scholarships to the students.
The funders had no role in study design, data collection and analysis, decision to publish, or preparation of the manuscript.

### Grant Disclosures

The following grant information was disclosed by the authors:
This work was supported by the Coordination for the Improvement of Higher Education Personnel (CAPES).
Brazilian National Council for Scientific and Technological Development (CNPq).
Rio de Janeiro Research Foundation (FAPERJ).
Funding Authority for Research and Projects (FINEP).

State University of North Fluminense "Darcy Ribeiro" (UENF) for granting scholarships to the students.

## Competing Interests

The authors declare that they have no competing interests.

## Author Contributions

- Tamires Cruz dos Santos conceived and designed the experiments, performed the experiments, analyzed the data, prepared figures and/or tables, authored or reviewed drafts of the paper, and approved the final draft.
- Mariana Ramos Leandro conceived and designed the experiments, performed the experiments, analyzed the data, prepared figures and/or tables, authored or reviewed drafts of the paper, and approved the final draft.
- Clara Yohana Maia conceived and designed the experiments, performed the experiments, prepared figures and/or tables, and approved the final draft.
- Patrícia Rangel performed the experiments, prepared figures and/or tables, and approved the final draft.
- Fabiano S. Soares performed the experiments, prepared figures and/or tables, and approved the final draft.
- Ricardo Reis performed the experiments, prepared figures and/or tables, and approved the final draft.
- Lucas Passamani performed the experiments, prepared figures and/or tables, and approved the final draft.
- Vanildo Silveira performed the experiments, prepared figures and/or tables, and approved the final draft.
- Gonçalo Apolinário de Souza Filho conceived and designed the experiments, analyzed the data, authored or reviewed drafts of the paper, and approved the final draft.

## Data Availability

Data is available at Zenodo: Tamires, Cruz dos Santos, Mariana, Ramos Leandro, & Gonçalo, Apolinário de Souza Filho. (2020). *Gluconacetobacter diazotrophicus* co-cultivated with *Arabidopsis thaliana* (Data set). Zenodo. DOI 10.5281/zenodo.3699504

## Supplemental Information

Supplemental information for this article can be found online at http://dx.doi.org/10.7717/peerj.9600#supplemental-information.

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
