# Peer review of "Arabidopsis thaliana exudates induce growth and proteomic changes in Gluconacetobacter diazotrophicus"

_PeerJ, doi:10.7717/peerj.9600_

## Round 0.1 · original submission · Major Revisions

Dear authors:

Thank you for considering PeerJ for your manuscript submission.

I have thoroughly reviewed your manuscript. In my judgment, the topic is highly interesting and state-of-the-art of genes in response of Gluconacetobacter diazotrophicus and its interaction with A. thaliana. However, a number of topics raised by the reviewers should be addressed to improve the technical quality of your manuscript prior to submission of a revised version of your MS.

When submitting your revised manuscript, please be sure the following points are addressed. Two reviewers raise substantial comments, that need to be addressed adequately.

Besides these major concerns, a number of more minor issues considering in the text also need to be addressed. I hope you will be able to address these issues in the next version of this interesting work.

Thank you for your progress, and we look forward to the revised revision.

·

Basic reporting

Cruz dos Santos and col. submitted the manuscript “EttA and AccC are essential for the response of Gluconacetobacter diazotrophicus to Arabidopsis thaliana exudates”, which deals with the physiological and proteomic characterization of G. diazotrophicus response to root exudates from A. thaliana. G. diazotrophicus PAI5 is a model bacterium for the study of the interactions between non-symbiotic diazotrophic bacteria and plants, in particular, monocot plants. Several aspects of the interaction with sugarcane, the natural host of G. diazotrophicus, are known. However, the growth promotion exerted by G. diazotrophicus on the model plant A. thaliana allows a deeper characterization of the interaction mechanisms that take place between plants and beneficial bacteria. This is the main focus of the manuscript and the main strenght of the results presented by the authors.
Cruz dos Santos and col. first analyzed the bacterial growth in a culture medium were A. thaliana was growing, finding that the bacterial growth was enhanced, and that the production of plant metabolites increasing the bacterial growth was constitutive. After analyzing the bacterial proteome, the authors found a group of proteins up- and down-regulated by the plant exudates. Finally, the authors compared the response to plant exudates of bacterial mutants in selected genes, reporting that the proteins EttA and AccC are essential for the growth increase of G. diazotrophicus.
The manuscript is clearly written and the well structured according to PeerJ standards.
For this Basic Reporting section of the review, in my opinion, several aspects should be revised.
-Manuscript title put the focused specifically in the results from the last experiment, while the main findings are the overall proteomic results. The title could be reformulated in this sense.
The Introduction section is very general. The authors could present specific information of what is already known about determinants from G. diazotrophicus involved in the interactions with plants.
Figures are relevant, though Figure 2 could be included as supplementary material.

Experimental design

Results presented in the manuscript are interesting and deserve attention, considering the importance of G. diazotrophicus as a model bacterium.
Materials and Methods are well described. A few changes could improve the understanding of the methodology and the results.
-Authors should provide details about the preparation of the bacterial inoculums [Lines (L.) 115-117]. Were the cells washed before inoculation to prevent carryover of spent medium? Were the cells starved? These are key questions for understanding the proteomic response, considering the chemical differences between DYGS medium and Murashige Skoog medium.
-In L. 113-116, it is not clear how many seedlings were utilized in each Petri dish.
-In the subsection 2.4.1, it is not clear the protocol for the protein extraction. According to L. 128-129, the authors did not follow standard protocols for cell disruption, treating the cell pellets directly with an “extraction buffer” composed of trichloroacetic acid, acetone and DTT. This solution would hardly allow a good extraction of intracellular proteins, which could explain why only 450 proteins were indentified (L. 269).
-The utilization of STRING database and the parameters utilized (including the type of interaction analyzed) should be mentioned in the “2.4.4 Proteomic data analysis” subsection (L. 184).

Validity of the findings

As stated before (see other sections of this review), the aims of the work and the results obtained by the authors are interesting and valuable.
In the Results and in the Discussion sections, the authors should consider several points that can highly improve the manuscript:
- How do the culture conditions influence the results and, in consequence, the conclusions? According to what is described in Materials and Methods section, G. diazotrophicus strains were first precultured in DYGS medium (which is composed of peptone and yeast extract, among others), and then utilized to inoculate MS medium (mainly composed of mineral salts). Culture conditions are key aspects to be considered when comparing the growth of the wild type strain and the mutants affected in protein synthesis, the pentose phosphate pathway and fatty acid synthesis.
-It is not clear the criteria to select the mutants to be analyzed.
-Two Glucose-6-phosphate 1-dehydrogenase homologs were identified as significantly up-regulated by LC/MS, which correspond to the Accessions A9H0G0 and A9H326 (a third homolog not identified by LC/MS is also present in the genome of G. diazotrophicus). Which one was utilized in the experiments? How does the presence of the second homolog influence the results? In addition, in L. 381-383 these two homologs are mentioned as Zwf and GDI_3177.
- Only few proteins were considered in the Discussion section.
- Besides, it is not clear why, for instance, RpsA and RplW are considered in the Discussion and not others also related to protein synthesis.
- L. 343-347. MetE is involved in the Activated Methyl Cycle and is related to the production of AI-2 quorum sensing (QS) molecule in different microorganisms. However, similar to the rest of the Alphaproteobacteria, G. diazotrophicus lacks a LuxS homolog required for AI-2 synthesis. In consequence, the up-accumulation of MetE should not be related to QS activity.
- L370-372. The authors considered that after 24 of cocultivation of G. diazotrophicus with A. thaliana, the bacterial cells were under energy depletion, which is difficult to consider since the cells seems to be still in exponential phase, according to Figure 1. In any case, if according to the literature EttA is related to energy depletion, would it be expectable to find this protein down-accumulated in the presence of the plant, since root exudates provide extra carbon and energy sources?
- As mentioned in L291-292, ClpX plays a role in protein degradation. However, ClpX is the ATP-binding subunit of the ClpXP protease, and in consequence does not show degradation activity on acyl homoserine lactones (L292-295). Interestingly, it has been shown that a mutation in the ClpX coding sequence produces an increase in acyl homoserine lactone production by Burkholderia cenocepacia (see Microbiol. Res. 2016, 186-187:90-98). However, to date it is not clear the mechanism.
-The authors should include in the Discussion section about AccC, the presence of a second accC homolog in the genome of G. diazotrophicus.

Additional comments

No comments.

·

Basic reporting

The paper entitled 'EttA and AccC are essential for the response of Gluconacetobacter diazotrophicus to Arabidopsis thaliana exudates' is very interesting. It explores the interaction between G. diazotrophicus and the model dicot Arabidopsis. The authors determine the bacterial translation response to the plant exudates. Their findings are very original.

Experimental design

My only concern is related to the complete demonstration of the participation of EttA and AccC in the bacterial response. I would ask the authors to either complement the mutants or resolve if the ettA and accC messengers do not include other transcripts or at least show in a map that those genes are not followed by other transcripts in operons.

Validity of the findings

The findings are almost validated, but it is necessary to be sure that only ettA and accC and not any other products are involved.

Additional comments

I have the additional following questions,

-were the seeds germinated in a liquid medium?

-why were the plantlets maintained under light all the time without darkness periods?

-Were the roots of the plantlets covered from the light?



Comments

I also would ask the authors to compare the published results of the G. diazotrophicus-sugarcane interaction with their results.

I would ask to discuss on transporters, like TonB which is also highly up-accumulated.

Line 216, "2.3" instead of "4.3"?

Line 223, "section 2.5" instead of "section 4.5"?

---

## Round 0.2 · accepted · Accept

Thank you for considering the Peer J for your manuscript submission titled “Arabidopsis thaliana exudates induce growth and proteomic changes in Gluconacetobacter diazotrophicus”.

Taking into account the comments of the two reviewers, I coincide with their opinions, the referees do acknowledge that the manuscript has merit, the manuscript was substantially improved taken into account the reviewer's recommendations.

Therefore, I recommend that the manuscript be accepted for publication; please attend to the suggested minor corrections.

Sincerely

·

Basic reporting

Tamires Cruz dos Santos, Gonçalo Apolinário de Souza Filho and col. submitted a revised version of the manuscript “Arabidopsis thaliana exudates induce growth and proteomic
changes in Gluconacetobacter diazotrophicus” (firstly entitled EttA and AccC are essential for the response of Gluconacetobacter diazotrophicus to Arabidopsis thaliana exudates).
I have carefully read the revised version and the corresponding letter with the point-by-point responses. I agree with all the changes the authors made on the manuscript, and I consider that it has been highly improved. I only have very minor comments that I include in the "General comments for the author" section.

Experimental design

No additional comments.

Validity of the findings

No additional comments.

Additional comments

Minor comments
- In Lines 104-105, the words “three times” are repeated.
- Use italics for A. thaliana in Lines 320, 322 and 413.
- Use lower case for “xylinus” in Line 334.
- Use upper case for Gram in Line 439.
- Check the references 18, 23, 24, 28, 37, 38, 47, 48, 49.

·

Basic reporting

The authors have achieved the main concerns. My opinion is that the manuscript has the quality to be accepted for publication.

Experimental design

No comment

Validity of the findings

No comment

Additional comments

I consider that the authors have included most of the demands.